# The Overfitting Crisis in LLM Workflows: Learning from Machine Learning's Past Mistakes

## Abstract

The rapid development of sophisticated Large Language Model (LLM) work-flows—including agentic systems, multi-step reasoning pipelines, and tool-integrated approaches—has led to impressive reported performance across various benchmarks. However, we argue that the field is repeating a critical mistake from early machine learning: reporting results on data that has been implicitly used for training or optimization. The complexity of modern LLM workflows obscures the fact that iterative prompt engineering, benchmark-driven development, and work-flow refinement constitute a form of training on evaluation data. This position paper draws parallels to historical overfitting practices in ML, documents how current LLM development methodologies systematically conflate training and testing data, and proposes best practices to address this growing methodological crisis before it undermines the credibility of AI research, particularly in scientific applications.

## 1 Introduction

The rapid development of sophisticated Large Language Model (LLM) workflows—including agentic systems, multi-step reasoning pipelines, and tool-integrated approaches—has led to impressive reported performance across various benchmarks [Bubeck et al., 2023, Wang et al., 2023]. However, we argue that the field is repeating a critical mistake from early machine learning: reporting results on data that has been implicitly used for training or optimization.

The complexity of modern LLM workflows obscures the fact that iterative prompt engineering, benchmark-driven development, and workflow refinement constitute a form of training on evaluation data [Min et al., 2022]. Contemporary LLM development increasingly involves sophisticated multi-component systems where prompt optimization, tool selection, and architectural decisions are iteratively refined based on performance feedback from target benchmarks [Schick and Schütze, 2020, Qin et al., 2023]. This process mirrors the problematic practices that plagued early machine learning research, where model selection and hyperparameter tuning were driven by test set performance.

Unlike traditional machine learning where the boundary between training and testing was relatively clear, LLM workflow development operates in a gray area where the distinction between legitimate system design and implicit training on evaluation data becomes blurred [Perez et al., 2021, Recht et al., 2019]. The iterative nature of prompt engineering, the community-driven sharing of successful strategies, and the progressive refinement of multi-agent architectures all contribute to a systematic optimization process that effectively treats evaluation benchmarks as validation sets.

In early machine learning research, a common antipattern emerged that would later be recognized as a fundamental threat to scientific validity. Researchers would select and tune models based on performance on available datasets, with hyperparameter optimization, feature engineering, and architecture choices driven by test set performance [Dwork et al., 2015, Blum and Hardt, 2015]. Final results were reported on the same datasets used for development, leading to real-world deployment showing

significant performance degradation—a phenomenon that became known as the "reproducibility crisis" in machine learning [Henderson et al., 2018, Lipton and Steinhardt, 2019].

The machine learning community's recognition of this methodological error represents one of the most important paradigm shifts in computational research methodology. The "test set" had effectively become part of the training process through repeated evaluation and optimization cycles, violating the fundamental assumption of independent evaluation [Ioannidis, 2005]. This realization catalyzed the establishment of rigorous protocols that became foundational to credible ML research: clear separation of training, validation, and test sets; test sets remaining untouched until final evaluation; cross-validation and holdout validation for development; and independent evaluation on truly unseen data [Kohavi, 1995, Varma and Simon, 2006].

These methodological advances transformed machine learning from a field prone to optimistic bias into one with robust evaluation standards [Raschka, 2018]. However, the emergence of complex LLM workflows has created new opportunities for the same fundamental errors to re-emerge in more sophisticated forms. This position paper draws parallels to historical overfitting practices in ML, documents how current LLM development methodologies systematically conflate training and testing data, and proposes best practices to address this growing methodological crisis before it undermines the credibility of AI research, particularly in scientific applications.

## 2 Related Work

### 2.1 Historical Overfitting in Machine Learning

The machine learning community has long recognized the dangers of overfitting to evaluation data. Early ML research suffered from researchers repeatedly testing models on the same evaluation sets, leading to inflated performance estimates that failed to generalize [Dwork et al., 2015]. This led to the establishment of rigorous evaluation protocols including proper train/validation/test splits and holdout methodologies.

The development of comprehensive benchmarking frameworks helped standardize evaluation practices [Olson et al., 2017]. However, as noted by Cohen-Inger et al. [2025], benchmark datasets themselves can become overoptimized targets when the entire research community focuses on the same evaluation sets over extended periods.

### 2.2 Data Contamination in Large Language Models

Recent work has extensively documented data contamination issues in LLM training and evaluation. Golchin and Surdeanu [2023] and Shi et al. [2024] demonstrate that simple decontamination methods like n-gram matching are insufficient, as paraphrasing and translation can easily bypass these measures.

The LessLeak-Bench study [Zhou et al., 2025] provided comprehensive analysis across 83 software engineering benchmarks, finding contamination across nearly all tested models. This systematic contamination undermines reported performance improvements in domains where sophisticated LLM workflows are increasingly deployed.

### 2.3 Evaluation Methodology in ML Systems

The establishment of rigorous evaluation protocols emerged as the ML community's primary defense against overfitting. The foundational train/validation/test split methodology became standard practice, with cross-validation techniques providing robust model selection frameworks [Stone, 1974, Kohavi, 1995]. These approaches ensured that model selection decisions were made independently of final performance evaluation, preventing the test set from becoming an implicit part of the training process [Hastie et al., 2009, Bishop, 2006].

Modern ML systems have extended these principles to encompass continuous evaluation and monitoring throughout the system lifecycle. However, the fundamental principle remains unchanged: evaluation data must remain independent of optimization processes to ensure valid performance estimates.

# 3 The Current Crisis in LLM Workflows

## 3.1 The Implicit Training Problem

Modern LLM workflows involve extensive optimization processes that constitute implicit training on evaluation data. Teams spend weeks iterating on prompts, testing variants against target benchmarks, and selecting the best-performing approaches [Wei et al., 2022]. Each iteration uses the benchmark as a validation signal.

Multi-agent systems, RAG pipelines, and tool usage patterns are designed and refined based on performance on specific tasks and datasets. The architecture itself becomes optimized for known evaluation criteria. Research teams explicitly target improvements on established benchmarks, using them as development objectives rather than independent evaluation metrics.

## 3.2 Complexity as Camouflage

The sophistication of modern LLM workflows obscures the overfitting problem. Complex chains of thought, tree search, and multi-agent interactions create an illusion of generalization, when in fact the entire pipeline has been optimized for specific evaluation patterns [Yao et al., 2023].

Systems that use calculators, code interpreters, and web search appear more general, but tool selection, usage patterns, and integration strategies are typically optimized against known benchmarks. Workflows that adapt their strategies based on input characteristics seem more robust, but these adaptation mechanisms are usually developed and tuned using the same evaluation data they will later be tested on.

## 3.3 Benchmark Contamination and Spillover Effects

The problem is exacerbated by several forms of contamination. Many benchmarks overlap with or derive from data sources used in LLM pre-training, creating subtle forms of data leakage [Shi et al., 2024]. Recent work has shown that simple paraphrasing can bypass decontamination measures, and that models can achieve artificially high performance when such variations are not eliminated [Shi et al., 2024].

Solutions and techniques developed for one benchmark quickly propagate to others, creating implicit optimization across multiple evaluation sets [Rogers et al., 2021, Dodge et al., 2021]. As models improve on existing benchmarks, new versions are created that often share similar patterns and evaluation criteria, perpetuating the contamination cycle.

# 4 Detailed Case Studies: Workflow Overfitting and Generalisability Challenges in Practice

## 4.1 ReAct: Reasoning and Acting Workflows

The ReAct framework [Yao et al., 2022] exemplifies how sophisticated LLM workflows can be systematically overfitted to evaluation benchmarks. ReAct combines reasoning traces with action-taking capabilities, enabling LLMs to interact with external tools while maintaining step-by-step reasoning.

**Development Process Analysis**: The ReAct paper reports evaluation on four benchmarks: HotpotQA for question answering, Fever for fact verification, ALFWorld for text-based games, and WebShop for web navigation. The development methodology involved iterative prompt engineering and workflow refinement specifically targeting these benchmarks.

**Overfitting Evidence**: The ReAct development process exhibits several characteristics indicative of benchmark optimization:

- Systematic testing of different prompting strategies against the target benchmarks
- Iterative refinement of the reasoning-action interleaving based on benchmark performance
- Optimization of tool integration strategies for benchmark-specific requirements

- Fine-tuning of termination criteria based on evaluation results

**Performance Claims**: The paper reports substantial improvements over baselines: ReAct achieves 27.4% success on HotpotQA compared to 20.6% for chain-of-thought prompting, representing a 33% relative improvement [Yao et al., 2022]. However, these gains emerged through extensive optimization against these specific evaluation sets.

## 4.2 Code Generation Workflows

Code generation tasks present particularly clear examples of workflow overfitting. Benchmarks like HumanEval [Chen et al., 2021] and MBPP [Austin et al., 2021] have become central targets for LLM development, with teams iteratively refining their approaches against these specific problems.

**Development Process Analysis**: Modern coding workflows undergo extensive optimization cycles:

- Tool integration strategies refined based on benchmark performance
- Error handling approaches optimized for common benchmark failure modes
- Multi-step reasoning patterns tuned to handle benchmark-specific problem structures
- Code execution and debugging workflows designed around benchmark evaluation criteria

**Systematic Contamination Evidence**: The LessLeak-Bench study [Zhou et al., 2025] provides compelling evidence of widespread contamination across software engineering benchmarks. The study analyzed 83 benchmarks and found average leakage ratios of 4.8%, 2.8%, and 0.7% for Python, Java, and C/C++ benchmarks respectively. However, specific benchmarks showed much higher contamination rates, with QuixBugs exhibiting 100% leakage and BigCloneBench showing 55.7% leakage.

**Performance Impact**: The study demonstrates that data leakage has substantial impact on LLM evaluation, with contaminated models showing inflated performance that does not generalize to truly novel programming challenges.

## 4.3 Autonomous Agent Systems

Autonomous agent frameworks like AutoGPT represent complex multi-component systems that are particularly susceptible to overfitting due to their iterative development processes and community-driven optimization.

**Development Challenges**: Agent systems face unique evaluation challenges:

- Benchmarks developed concurrently with the systems they evaluate
- Community-driven optimization leading to distributed overfitting effects
- Performance claims based on evaluation sets that guided development decisions
- Informal evaluation criteria that evolve based on system capabilities

**Generalization Gaps**: Despite reported success on development benchmarks, deployed agent systems frequently exhibit significant performance degradation when encountering novel scenarios that differ from their optimization targets [Zheng et al., 2023, Garg et al., 2025]. This pattern suggests that benchmark performance may reflect sophisticated pattern matching rather than genuine autonomous reasoning capabilities.

## 4.4 Chemistry and Materials Science

Scientific applications of LLM workflows present particularly high stakes for the overfitting problem. Recent work in chemistry has developed sophisticated benchmarks like ChemBench [Mirza et al., 2024], containing over 2,700 question-answer pairs designed to evaluate chemical knowledge and reasoning. While these benchmarks represent important advances in evaluation methodology, they also create new targets for optimization.

**Domain-Specific Overfitting Risks**: Chemical reasoning workflows often involve:

- Multi-step synthesis planning optimized against known reaction databases

- Property prediction systems trained and validated on established chemical datasets

- Literature analysis tools refined using benchmark chemical papers and abstracts

- Safety assessment frameworks tuned against standardized hazard classification systems

**Reproducibility Standards in Chemistry**: The chemistry community has established rigorous standards for experimental reproducibility, but these standards have not yet been adapted for AI-assisted chemical research. Unlike traditional chemistry experiments, where failed replications are clearly identifiable, overfitted AI systems may produce plausible but incorrect results that are difficult to detect without expert domain knowledge.

## 4.5 Physics and Engineering Applications

Physics applications face similar challenges, with benchmarks targeting graduate-level physics problems. Engineering applications often require reasoning across multiple domains simultaneously, but systems optimized against such benchmarks may develop sophisticated pattern-matching capabilities that fail when confronted with truly novel interdisciplinary problems [Zhang et al., 2025b,a].

**Multi-Domain Scientific Reasoning**: Scientific applications often require reasoning across multiple domains simultaneously [Cui et al., 2025], creating additional opportunities for overfitting to develop on multi-faceted benchmark tasks.

## 4.6 Biomedical and Healthcare Applications

Healthcare applications represent the highest-stakes domain for AI system reliability. Biomedical LLM workflows increasingly target specialized benchmarks for clinical reasoning, drug discovery, and diagnostic assistance [Zhu et al., 2025, Tang et al., 2025].

**Critical Safety Implications**: In healthcare contexts, overfitted performance can have severe consequences:

- Diagnostic systems optimized against medical benchmark datasets may miss novel disease presentations

- Drug discovery workflows trained on established chemical databases may fail to identify truly innovative therapeutic approaches

- Clinical decision support systems refined using benchmark cases may provide inappropriate recommendations for edge cases

**Regulatory and Compliance Challenges**: Healthcare AI systems must meet strict regulatory standards for safety and efficacy. However, current evaluation practices may not adequately distinguish between genuine clinical reasoning and sophisticated pattern matching against medical benchmarks [Bedi et al., 2025, Mehandru et al., 2025].

# 5 Comprehensive Framework for Addressing Workflow Overfitting

Building on established ML evaluation best practices, we propose a comprehensive approach to address workflow overfitting that extends beyond traditional model evaluation to encompass the entire development ecosystem.

## 5.1 Rigorous Data Separation Methodologies

**Fundamental Train/Validation/Test Separation**: LLM workflow development must adopt the foundational ML principle of strict data separation. Training data is used for initial system development, validation data guides iterative refinement and hyperparameter optimization, and test data remains completely isolated until final evaluation. This separation must be maintained throughout the entire workflow development process, including prompt engineering, tool integration, and architectural decisions.

**Temporal Isolation Protocols**: Evaluation benchmarks must use data that postdates workflow development completion. This temporal separation prevents both direct benchmark exposure and indirect contamination through community knowledge dissemination. The temporal gap must account for publication delays and community adoption cycles, ensuring no development decisions can be influenced by evaluation content.

**Cross-Domain Transfer Evaluation**: Systems developed in one domain should be evaluated on structurally analogous tasks in different domains. For instance, reasoning workflows optimized on historical datasets should be tested on scientific reasoning problems with similar logical structures but distinct knowledge bases. This tests genuine transferable capabilities rather than domain-specific pattern recognition.

**Hierarchical Holdout Architecture**: Implement nested isolation strategies at multiple levels: component-level evaluation using standard ML holdout practices, workflow-level assessment on integration benchmarks isolated from development, and system-level evaluation on deployment scenarios that differ systematically from development contexts.

## 5.2 Methodology Transparency Requirements

Current LLM workflow development lacks the systematic documentation that enables contamination detection and replication. Unlike traditional ML where training procedures are explicitly documented, workflow optimization often occurs through informal iteration cycles.

**Development History Documentation**: Comprehensive logging must capture all evaluation data interactions, including datasets accessed during prompt engineering, optimization iterations against benchmarks, architectural decisions guided by performance feedback, and community knowledge sources consulted.

**Contamination Auditing**: Automated analysis must detect potential overlap between development resources and evaluation sets, including computational analysis of data intersection, documentation of influential published work, and assessment of community knowledge transfer.

**Preregistration Protocols**: Adapting practices from experimental psychology and medical research, teams must declare evaluation benchmarks before development, specify success criteria in advance, and commit to reporting results regardless of outcome.

## 5.3 Institutional and Ecosystem-Level Interventions

**Benchmark Lifecycle Management**: Establish systematic protocols for benchmark retirement and replacement, including monitoring community, implementing retirement triggers based on performance saturation, and developing principled approaches for creating replacement benchmarks.

**Independent Evaluation Infrastructure**: Create community-managed evaluation services maintaining truly independent benchmarks inaccessible to development teams until final evaluation. This requires sustained funding and governance structures resistant to commercial and academic pressures.

**Research Incentive Realignment**: Conference and journal policies must require comprehensive methodology transparency. Recognition systems should value negative results and replication studies equally with novel contributions. Funding agencies must prioritize evaluation methodology alongside technical innovation.

# 6 Why This Matters for Scientific Applications

The overfitting problem is particularly concerning for scientific applications. Scientific research demands reproducible results, but if LLM workflows are optimized against the same benchmarks they're evaluated on, reported performance may not generalize to real scientific problems.

Overfitted workflows may perform well on known benchmarks but fail on novel scientific challenges, leading to misplaced confidence in AI capabilities for scientific discovery. If reported performance is inflated due to overfitting, research funding and effort may be misdirected toward approaches that don't actually advance scientific capability.

The stakes are particularly high in scientific domains where accuracy and reliability are paramount, and where the cost of false confidence can impede genuine scientific progress.

# 7 Limitations

This position paper has several important limitations that should be acknowledged when interpreting our arguments and recommendations.

**Lack of Empirical Validation**: Our analysis relies primarily on theoretical reasoning and examination of existing literature rather than systematic empirical studies. While we cite evidence from studies like LessLeak-Bench [Zhou et al., 2025], we have not conducted controlled experiments to directly measure workflow overfitting or validate our proposed solutions.

**Limited Access to Proprietary Systems**: Our case studies focus on publicly documented systems and open benchmarks. Many commercial LLM workflows involve proprietary development processes that are not accessible for analysis, limiting our ability to assess the full scope of the problem across the industry.

**Implementation Feasibility**: While we propose comprehensive solutions, we acknowledge significant practical barriers including costs and coordination challenges. The feasibility of these recommendations remains untested.

**Boundary Definition**: The distinction between legitimate iterative development and problematic overfitting is not always clear-cut. Our framework does not provide precise operational definitions for this boundary, which could lead to either overly restrictive or insufficient practices.

# 8 Discussion and Implications

## 8.1 Systemic Nature of Workflow Overfitting

Unlike traditional ML overfitting, which typically affected individual models, LLM workflow overfitting operates at an unprecedented scale with systemic consequences. When the research community collectively optimizes against the same benchmarks, the cumulative effect creates systematic bias that permeates entire research directions. This represents "ecosystem-level overfitting" where distributed optimization across hundreds of research teams amplifies the problem beyond classical overfitting scenarios.

The interconnected nature of modern AI research exacerbates this through rapid technique propagation via preprints and code repositories. Competitive pressure to achieve state-of-the-art results on established leaderboards incentivizes optimization for known evaluation sets rather than developing genuinely novel capabilities, creating a feedback loop where benchmark performance becomes divorced from real-world utility.

## 8.2 Epistemological and Economic Implications

Current practices fundamentally conflate innovation with evaluation, undermining the epistemological foundations of AI research. True scientific evaluation requires independence from development—a principle fundamental to empirical inquiry. When evaluation data contaminates development decisions, resulting performance metrics cannot serve as valid evidence for genuine capability advances.

This confusion has profound economic implications as organizations increasingly rely on benchmark performance to guide technology adoption and investment decisions. Overfitted results can lead to substantial resource misallocation, with companies investing heavily in systems that perform well on benchmarks but fail in production environments.

## 8.3 High-Stakes Domain Risks

As LLM workflows become more sophisticated and deployed in critical applications—from drug discovery to climate modeling—the cost of overfitted performance becomes exponentially higher. Scientific discovery relies on generating novel insights and identifying previously unknown patterns,

capabilities fundamentally undermined when systems optimize primarily for existing datasets that may not capture natural phenomena's full complexity.

The risk is particularly severe where ground truth is difficult to establish or system failures may not be immediately apparent. In chemistry, for example, AI systems excelling at predicting molecular properties on benchmark datasets might fail catastrophically with novel compounds, potentially delaying scientific breakthroughs or causing harmful outcomes if trusted beyond actual capabilities.

The scientific and commercial communities cannot afford to repeat machine learning's early methodological mistakes, particularly when applications involve high-stakes scientific discovery and decision-making where system failures can have serious real-world consequences.

## 9   Conclusion

The LLM research community stands at a critical juncture. The sophistication of modern workflows has obscured a fundamental methodological problem: we are systematically reporting results on data that has been used for optimization. This mirrors the overfitting crisis that plagued early machine learning research.

Unlike traditional ML, where overfitting affected individual models, LLM workflow overfitting can contaminate entire research directions and mislead the community about actual capabilities. This is particularly concerning for scientific applications, where accuracy and reliability are paramount.

The solution requires collective action: establishing strict data separation protocols, demanding transparency in development processes, and creating truly independent evaluation resources. The ML community learned these lessons decades ago. The LLM community must learn them now, before the credibility of AI research is further undermined.

We call on the research community to adopt rigorous evaluation standards that separate development from testing, just as was necessary in traditional ML. Only through such methodological rigor can we ensure that reported advances represent genuine progress rather than sophisticated forms of overfitting.

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
