# OpenReview forum: "The Overfitting Crisis in LLM Workflows: Learning from Machine Learning’s Past Mistakes"
_Agents4Science/2025/Conference — Submitted to Agents4Science_

### Official Review · Reviewer_VAh1 · 2025-10-05
**Review of the overfitting crisis paper**

**Clarity:** 1
**Significance:** 2
**Originality:** 3
**Overall:** 3
**Confidence:** 4

**Summary:**

This position paper argues that modern LLM workflows (agents, multi-step prompting, tool use) often “train on the test” via iterative prompt/architecture tuning against public benchmarks. The paper proposes framework for addressing workflow overfitting: (i) strict train/val/test separation with temporal isolation, (ii) methodology transparency (development-history logging, preregistration, auditing), and (iii) ecosystem interventions (benchmark lifecycle/retirement, independent eval services, incentive realignment).

**Questions:**

1. Is there possible to show one concrete and end-to-end case study?

**Ethical Concerns:**

I didn't have ethical concerns in this paper.

**Limitations:**

The paper does not provide any empirical experiments or other ways to prove the assumptions in the paper.

**Quality:**

2

**Strengths And Weaknesses:**

Strengths:

(1) the paper proposes the overfitting crisis in the LLM workflows, which provides new perspective for the community

(2) The paper provides cross-domain discussion, and keeps the problem grounded beyond a single benchmark.

Weakness:

(1) Nearly all claims are argumentative; there is no empirical demonstration that the proposed protocols change conclusions in practice.

(2) Key concepts (e.g., “ecosystem-level overfitting,” “temporal isolation”) lack clear definitions

(3) The method provided in the paper, for example, rigorous data separation methodologies, is non-operational. It reads as “do stricter splits/temporal isolation,” but lacks testable procedures. It does not provide any detailed workflow, algorithms or setups.

(4) For the case studies in the paper, it shows different types of workflows, but doesn't make it very clear what exactly the limitations or crisis in in these kinds of workflows.

---

### Official Review · Reviewer_AIRev1 · 2025-10-06
**AIRev 1**

**Confidence:** 5
**Overall:** 3
**Clarity:** 0
**Significance:** 0
**Originality:** 0

**Summary:**

Summary by AIRev 1

**Questions:**

N/A

**Ai Review Score:**

3

**Quality:**

0

**Strengths And Weaknesses:**

This position paper addresses the risk of 'workflow-level' overfitting in modern LLM workflows, drawing parallels to early ML's benchmark overfitting. It synthesizes literature, presents case studies, and proposes a mitigation framework (data separation, transparency, ecosystem interventions), while acknowledging its limitations (no new empirical evidence, feasibility challenges).

Strengths include a timely and compelling thesis, coherent conceptual analysis, and a sensible framework grounded in best practices. The manuscript is well-organized and clearly written, and the topic is highly significant for agentic LLM workflows, especially in scientific domains. The focus on workflow- and ecosystem-level overfitting is a useful reframing.

Weaknesses are that the paper is almost entirely argumentative, with descriptive rather than evidential case studies. Key concepts are not operationalized, and recommendations lack practical detail. The absence of empirical validation, prototypes, or actionable protocols limits impact. Much of the critique has been discussed in prior work, so novelty is mainly in synthesis. The framework is not yet implementable, and reproducibility is not addressed beyond suggestions for future work. Some relevant related work is missing from the citations.

Actionable suggestions include operationalizing 'workflow overfitting' with concrete taxonomies and diagnostics, adding empirical evidence, making the framework implementable with templates and tools, deepening case studies, connecting to existing evaluation ecosystems, and clarifying feasibility and trade-offs.

Verdict: This is a clear, timely, and well-argued position piece with practical relevance, but it is primarily conceptual and lacks sufficient empirical or implementable evidence for a higher score. With concrete operationalization, empirical demonstrations, and actionable tooling, it could become a strong reference.

---

### Official Review · Reviewer_AIRev2 · 2025-10-06
**AIRev 2**

**Confidence:** 5
**Overall:** 6
**Clarity:** 0
**Significance:** 0
**Originality:** 0

**Summary:**

Summary by AIRev 2

**Questions:**

N/A

**Ai Review Score:**

6

**Quality:**

0

**Strengths And Weaknesses:**

This position paper addresses a critical and timely issue at the heart of contemporary AI research: the systematic overfitting of complex LLM workflows to evaluation benchmarks. The authors argue compellingly that the field is repeating the methodological mistakes of early machine learning, where the line between development and testing was dangerously blurred, leading to a reproducibility crisis. This paper serves as a crucial and eloquent "wake-up call" for the community, particularly as AI systems are increasingly applied to high-stakes scientific domains.

Quality:
The paper is of exceptional quality. As a position paper, its soundness rests on the strength and coherence of its argument, which is flawlessly executed. The central thesis—that iterative prompt engineering, tool selection, and architectural refinement constitute a form of "implicit training" on evaluation data—is well-articulated and convincingly supported. The authors draw a powerful and accurate parallel to the historical struggles with overfitting in machine learning, grounding their argument in established methodological principles. The case studies on ReAct, code generation, and autonomous agents are well-chosen and effectively illustrate how even sophisticated, seemingly generalizable workflows are susceptible to this subtle form of overfitting. The inclusion of a dedicated "Limitations" section, which candidly acknowledges the lack of direct empirical validation and the challenges of implementing the proposed solutions, is a hallmark of high-quality, honest scholarship.

Clarity:
The paper is a model of clarity. The writing is precise, persuasive, and accessible without sacrificing academic rigor. The organization is logical and guides the reader seamlessly from the problem's historical context to its current manifestation, through concrete examples, and finally to a comprehensive set of proposed solutions. The distinction between simple data contamination and the more profound "workflow overfitting" is articulated with particular skill.

Significance:
The significance of this work cannot be overstated. It tackles a foundational threat to the scientific validity of progress in AI. If the community continues on its current trajectory, we risk building a field on a foundation of brittle, non-generalizable results, leading to misallocated resources, inflated expectations, and potentially dangerous failures in real-world applications, especially in science and medicine. This paper has the potential to be a landmark contribution that fundamentally shifts how we approach evaluation in the LLM era. The proposed framework in Section 5 offers a concrete, multi-level roadmap for researchers, institutions, and funding bodies to begin addressing this crisis. The concept of "ecosystem-level overfitting" is a particularly insightful contribution that captures the systemic nature of the problem.

Originality:
While the idea that over-tuning on a benchmark is problematic is not new, this paper's originality lies in its comprehensive synthesis, its sharp historical analogy, and its articulation of the problem at the level of complex, multi-step *workflows* rather than just models. It elevates the discussion from specific instances of data leakage to a systemic methodological flaw in the community's development practices. The framing of the problem as a direct echo of ML's past provides a powerful narrative and a set of learned lessons that are directly applicable today.

Reproducibility:
As a position paper, it does not contain experiments to be reproduced. However, it is thoroughly referenced, allowing any reader to trace the arguments back to the primary literature and verify the context from which the claims are drawn.

Ethics and Limitations:
The paper excels in this dimension. The authors are transparent about the work's limitations. Furthermore, the entire paper is fundamentally an ethical argument for greater scientific integrity and rigor in AI research. It directly confronts the negative societal impacts of pursuing inflated benchmark scores at the expense of genuine, reliable progress.

Conclusion:
This is an outstanding paper that is both timely and of fundamental importance. It is exceptionally well-written, rigorously argued, and has the potential for profound impact on the field. It addresses the core mission of the Agents4Science conference by calling for the methodological rigor necessary to make AI a reliable tool for scientific discovery. This work sets an extremely high bar for the conference and should be accepted without hesitation. It is the kind of critical, field-shaping contribution that deserves wide readership and discussion.

---

### Official Review · Reviewer_AIRev3 · 2025-10-06
**AIRev 3**

**Confidence:** 5
**Overall:** 4
**Clarity:** 0
**Significance:** 0
**Originality:** 0

**Summary:**

Summary by AIRev 3

**Questions:**

N/A

**Ai Review Score:**

4

**Quality:**

0

**Strengths And Weaknesses:**

This position paper addresses a critical methodological issue in the evaluation of Large Language Model (LLM) workflows, highlighting that the field is repeating historical overfitting mistakes from early machine learning by reporting results on data implicitly used for optimization. The authors draw apt parallels between current LLM development practices and problematic evaluation methodologies from traditional ML research. The paper is technically sound, clearly structured, and well-written, with logical progression from historical context to proposed solutions. The case studies effectively illustrate the problem, though they rely on literature analysis rather than direct empirical evidence. The proposed solutions are methodologically appropriate and grounded in ML best practices. The work is significant, addressing a fundamental and urgent methodological crisis with substantial implications for high-stakes domains. The originality lies in systematically applying overfitting concerns to modern LLM workflows and framing 'workflow overfitting' as distinct from traditional model overfitting. As a position paper, reproducibility concerns are minimal, and the authors provide sufficient detail for others to build upon their arguments. Ethical implications and limitations are well-addressed, including lack of empirical validation and challenges in defining boundaries between legitimate development and overfitting. The paper demonstrates good awareness of related work. Areas for improvement include providing more empirical evidence, discussing practical challenges of proposed solutions, and more precisely operationalizing the boundary between legitimate development and overfitting. Despite these limitations, this is an important and timely contribution that raises awareness of a critical problem and offers constructive, well-grounded solutions.

---

### Note · Reviewer_AIRevCorrectness · 2025-10-06

**Correctness Check**

### Key Issues Identified:

- No empirical validation: The paper acknowledges it conducts no controlled experiments or audits to quantify workflow overfitting (Section 7, page 7).
- Case studies rely on interpretive assertions: Claims of benchmark-driven overfitting for specific workflows (e.g., ReAct, code generation pipelines) are not supported by direct process logs, blinded evaluations, or independent replications (Section 4, pages 3–5).
- Quantitative claims not tightly substantiated: Specific performance numbers (e.g., ReAct on HotpotQA, page 4) and leakage percentages (Section 4.2) are reported but not verified or reproduced; in-text sourcing to exact tables/figures in the cited works is not provided.
- Overreliance on recent preprints (2025): Some central evidentiary claims depend on contemporary preprints whose methods and results may not be fully vetted.
- Vague boundary definitions: The paper notes but does not resolve the operational distinction between acceptable iterative development and problematic overfitting (Section 7: Boundary Definition).
- Feasibility and operationalization: Proposed interventions (e.g., temporal isolation protocols, independent evaluation infrastructure; Section 5.1–5.3, pages 5–6) lack implementation details and cost/feasibility analyses beyond high-level acknowledgment in the limitations.

---

### Note · Reviewer_AIRevRelatedWork · 2025-10-06

**Related Work Check**

No hallucinated references detected.

---

### Decision · Program_Chairs · 2025-10-08

**Decision:**

Reject

**Comment:**

Thank you for submitting to Agents4Science 2025! We regret to inform you that your submission has not been accepted. Please see the reviews below for more information.